# Characteristics and Trends of COVID-19 Infection in a Tertiary Hospital in Romania: A Retrospective Study

**DOI:** 10.3390/jpm12111928

**Published:** 2022-11-18

**Authors:** Isabela Ioana Loghin, Ioana Florina Mihai, Manuel Florin Roşu, Iulia Elena Diaconu, Andrei Vâţă, Radu Popa, Mihaela Cătălina Luca

**Affiliations:** 1Department of Infectious Diseases, “Grigore T. Popa” University of Medicine and Pharmacy, 700115 Iasi, Romania; 2Department of Infectious Diseases, “St. Parascheva” Clinical Hospital of Infectious Diseases, 700116 Iasi, Romania; 3Equilibrum Clinic, 700142 Iasi, Romania; 4Vascular Surgery Department, “St. Spiridon” Emergency Clinical Hospital, 700111 Iasi, Romania

**Keywords:** COVID-19, severe forms, comorbidities, antiviral therapy

## Abstract

(1) Background: The outbreak of the COVID-19 pandemic represented a real challenge for all of humanity. Characterized by a complex spectrum of signs and symptoms, by various severity degrees, the disease spread rapidly around the world. After more than two and half years since the beginning of COVID-19 pandemic, in the context of a paradoxical, enigmatic, and relentless COVID-19, the objective of the current study was to evaluate the characteristics and evolution of patients with SARS-CoV-2 infection, hospitalized in “St. Parascheva” Clinical Hospital of Infectious Diseases (Iasi, Romania). (2) Methods: This is a retrospective study that used the medical database recorded between July and November 2021 in order to highlight the characteristics of SARS-CoV-2 infection in patients from the northeastern region of Romania. (3) Results: We enrolled in the study a total of 1732 SARS-CoV-2 infected patients, mean age 67 ± 3.4 years, the female gender predominating (987 cases; 56.98%) as well as patients from the urban environment (982 patients; 56.69%). Moderate form of the disease predominated (814 cases; 47%), pulmonary imaging changes were found in 1042 (60.16%) cases, and 1242 (71.71%) patients had at least one underlying disease. After a median length of hospitalization of 9.5 days, 1359 (78.46%) patients were discharged cured, 48 (2.77%) were transferred to other services by decompensating the associated pathologies, 302 (17.43%) patients needed extensive support in the intensive care unit and there were 325 (18.76%) deaths. (4) Conclusions: The epidemiological characteristics of SARS-CoV-2 infection recorded in our study were mostly the same as characteristics of COVID-19 from all over the world.

## 1. Introduction

The end of 2019 brought to the fore a new coronavirus that infects humans, called SARS-CoV-2, that emerged in Wuhan and rapidly spread to other regions of China and around the world [1]. COVID-19, the name of the disease caused by this pathogen, is described as a complex spectrum of signs and symptoms, such as cough, shortness of breath, sore throat, in association with fever, also adding anosmia, ageusia, and gastrointestinal symptoms such as nausea, vomiting, and diarrhea, by various severity degrees [2].

On 11 March 2020, the World Health Organization (WHO) characterized COVID-19 as a pandemic disease and public health threat of international interest [3], in the context of the excessive increase in the number of COVID-19 confirmed cases with a wide geographical distribution [4]. 

Regarding the appearance of COVID-19 in Romania, the first case was confirmed on 26 February 2020 in county Gorj, of a person who was in close contact with a citizen from Italy, the most endemic area at that time [5]. Since then, until the time when this article was written (3 September 2022) in Romania, there have been 3,224,382 cases of COVID-19 and 66,751 deaths with a total number of 609,584,678 cases worldwide and 6,501,648 deaths globally [6].

Regarding pathogenesis, the SARS-CoV-2 enters the human body being transmitted by a respiratory way and infects human cells. Later, the infection stage is reached by binding to the cell surface protein angiotensin-converting enzyme 2 (ACE2) through the Receptor Binding Domain (RBD) of its spike (S) protein. Finally, it infiltrates and circulates in the cells of the immune system. In addition, the cellular transmembrane serine protease 2 (TMPRSS2) is required for the priming of the virus S protein [7,8]. At the epithelial cells, a complex mechanism can occur through the invasion of SARS-CoV-2 with ACE-2 receptors such as renal distal tubules, the hepatocytes, pancreas cell, and the mucosa of the intestine [9]. At the same time, noble organs can be affected, such as the brain, heart, and spleen, but also the lymphatic system (lymph nodes and other lymphoid tissues) [10].

Since the beginning the COVID-19 pandemic, numerous mutations of the SARS-CoV-2 have been identified, but it turns out that only a few managed to modify the immune features of the virus. The most significant variants influencing the function of the virus originated in the United Kingdom, South Africa, Brazil, and India [11]. They were named using the letters of the Greek alphabet, namely Alpha (B. 1. 1. 7), Beta (B. 1. 351), Gamma (P. 1), Delta (B. 1. 617. 2), and Omicron (B. 1. 1. 529) [12].

The Alpha strain (B1.1.7 or British variant) was discovered In September 2020 and represents one of the first distinct variants. The second variant of the coronavirus, the Beta variant (B.1.351), was detected in South Africa on 20 May 2020. The Beta variant was also the first to generate discussion about the effectiveness of vaccines against virus mutations. The Gamma strain, the third variant of the coronavirus, was detected in Brazil in November 2020. The fourth variant of the coronavirus was detected in India in October 2020. Called Delta, it is the most virulent strain, which has generated a new wave pandemic despite the fact that in most of the developed countries the vaccination process had begun. The last strain, Omicron (B. 1.1. 529), was described in November 2021 in South Africa [11,13].

All these strains represent mutations of the SARS-CoV-2 from the beginning of the pandemic until now. Some of these, due to their increased transmissibility, high mortality, or hospitalization rates, have been called variants of concern [14].

The Delta variant is one of the variants of concern strain of the SARS-CoV-2. Globally it is considered the determining agent of the most severe wave of diseases. At the same time, the Delta variant caused the highest number of deaths of COVID-19, among all other strains [15].

More than two and half years after the pandemic, we have learned that the clinical picture can range from asymptomatic forms or forms with mild or moderate symptoms to severe or critical cases [16]. The clinical features include a wide variety of manifestations, from respiratory symptoms, digestive symptoms, or skin rashes, to neurological damage [17]. Studies have shown that some groups of patients were treated at home or they were outpatients and others who developed a moderate to severe illness requiring mechanical ventilation were hospitalized in intensive care departments. A high mortality was recorded in cases which associated multiorgan dysfunction syndrome [18]. SARS-CoV-2 has affected infants, adolescents, and adults, people without comorbidities or with various underlying diseases [19]. The need to discover a vaccine which can induce SARS-CoV-2 specific neutralizing antibodies was the perfect and imperative solution in the absence of a specific treatment targeting life cycle of SARS-CoV-2 [20].

During the pandemic, great efforts were made to establish a national protocol for follow-up, diagnosis, and treatment of patients with SARS-CoV-2 infection. The research teams, doctors, and health and governmental institutions updated and made decisions based on the increased number of cases in each wave, as well as the identification of the predominant circulating strain, all this to ensure the good health of population in each territory, to keep the situation under control, to inform citizens, in order to remove some panic situations. Moreover, specific vaccines were launched, which were administered in a controlled manner. At the same time, SARS-CoV-2 testing was instituted in large communities with medical activity, in school communities, or among tourists.

In the context of a paradoxical, enigmatic, and relentless COVID-19, the objective of the current study was to evaluate the characteristics and the evolution of patients with SARS-CoV-2 infection, hospitalized in “St. Parascheva” Clinical Hospital of Infectious Diseases (Iasi, Romania).

## 2. Materials and Methods

### 2.1. Studied Patients

We conducted a retrospective study using the medical database recorded between July and November 2021 in order to assess the characteristics and evolution of SARS-CoV-2 infection in patients belonging to the northeastern region of Romania. In our study the inclusion criteria represented all in-patients confirmed with COVID-19 by reverse transcriptase-polymerase chain reaction (RT-PCR), hospitalized in the “St. Parascheva” Clinical Hospital of Infectious Diseases, Iasi in the previously mentioned period. The exclusion criteria in our study were patients that had a negative result in RT-PCR for SARS-CoV-2 infection.

### 2.2. Data Collection

The analysis included demographic data, medical history, clinical, laboratory and imagistic findings, the treatment administered, and patient’s evolution. The positive diagnostic was established after RT-PCR tests performed in our hospital molecular biology laboratory or in approved accredited laboratories from Iasi, Romania.

Study limitations: this is a retrospective study based on medical records, laboratory, and imagistic findings; furthermore, the identification of circulating variants was not possible in all cases, due to limitations related to the availability of RT-q PCR kits for detection of SARS-CoV-2 in the molecular biology laboratory of the “St. Parascheva” Clinical Hospital of Infectious Diseases, Iasi.

## 3. Results

During July to November 2021, a total of 1732 SARS-CoV-2 infected patients with complete dataset were admitted at “St. Parascheva” Clinical Hospital of Infectious Diseases in Iasi.

### 3.1. Patients Characteristics

The demographic characteristics of the patients (Table 1) show that the mean age was 67 ± 3.4 years, most of the patients affected being aged 65–74 years old (474 cases; 27.36%). Predominantly, the patients in our study were over the age of 50, having associated multiple comorbidities with deficient immunity, this age group being likely to develop moderate-severe forms of COVID-19. The lower percentage of patients (16.95%) in our study was represented by group age of young adults (18–44 years old) and pediatric ones (under 18 years), which we correlated with a better immune status, in the absence of an underlying disease. Feminine gender was most affected, compared with the masculine gender (987 cases; 56.99% vs. 745 cases; 43.01%). Most cases were from urban areas (982 patients; 56.70%) compared to rural areas (750 patients; 43.30%).

Of the 1732 hospitalized patients, 1242 (71.71%) patients had at least one underlying disease, such as cardiovascular pathology: hypertension (527 cases; 30.42%), atrial fibrillation (62 cases; 3.57%), heart failure (105 cases; 6.06%), chronic peripheral venous insufficiency (56 cases; 3.32%); metabolic pathology: diabetes (285 cases; 16.45%), mostly type 2, obesity (343 cases; 19.80%), chronic kidney disease (78 cases; 4.50%); but also oncological pathology (87 cases; 5.02%) (Figure 1).

### 3.2. Clinical Findings

The first clinical symptoms at the majority of patients with SARS-CoV-2 infection were: physical asthenia (99.88%) in varying degrees, fever (1230 cases; 71.01%), cough (982 cases; 56.69%), chest pain (870 cases; 50.23%), sore throat (930 cases; 53.69%), but also myalgia (675 cases; 38.98%), arthralgia (564 cases; 32.56%), anosmia (431 cases; 24.88%), ageusia (232 cases; 13.39%), abdominal pain (482 cases; 27.82%), diarrhea (387 cases; 22.34%), and loss of appetite (1110 cases; 64.08%), (see Figure 2), with an average duration from onset to hospitalization of 7 ± 3 days.

Regarding the clinical stage of the disease, in our study, the moderate form predominated (814 cases; 47%). There were also 669 severe forms (38.63%) and 249 mild forms (14.38%).

### 3.3. Investigations

Each patient in the study was examined paraclinically (hematological, biochemical, serological) and by imaging (computer tomography or chest radiography). The most frequent anomalies that we could observe in this study were the decreased number of white blood cells and lymphopenia, elevated liver-function values and inflammatory markers (CRP and LDH levels) and thrombocytopenia.

During the period of our study, in Romania, the dominant spreading variant was Delta. Therefore, the sequencing was performed especially in severe cases, the Delta variant (B. 1. 617. 2) being identified by RT-PCR in 282 patients.

The necessary imaging investigations included, for each patient, lung radiography or chest computer tomography. The most pulmonary imaging changes were found in 1042 (60.16%) cases, such as viral pneumonia 850 (49.07%) cases, interstitial pneumonia 138 (7.97%) cases, bronchopneumonia 38 (2.20%) cases, and 18 (1.03%) cases of lobar pneumonia.

### 3.4. Treatment and Evolution

Biological therapy with tocilizumab and anakinra, antivirals such as remdesivir and favipiravir, and new monoclonal antibody therapies were used as treatment for these patients. Most of them received antivirals (56.76%), biological therapy (20.09%), monoclonal antibody therapy (0.58%), and in 8.66% of cases were used both antivirals and biologic therapy (Figure 3).

In all cases, pathogenic and symptomatic treatment was associated. Most received anticoagulant treatment according to current guidelines and intravenous rebalancing infusions. In the case of patients with comorbidities, chronic treatment was also administered during hospitalization, and patients with acute respiratory failure needed oxygen therapy.

The median length of hospitalization was 9.5 days; 1359 (78.46%) patients were discharged cured, 48 (2.77%) were transferred to other services by decompensating the associated pathologies, 302 (17.43%) patients needed extensive support in the intensive care unit, and there were 325 (18.76%) deaths (Figure 4).

The most frequent deaths were recorded in the 65–74 age group (30.15%) and over 75 years old (45.84%), predominantly among male patients (55.69%) from urban area (63.38%), (Table 2). The underlying diseases often registered in the death’s cases were hypertension (41.84%), obesity (16.30%), and diabetes mellitus (14.46%). The applied treatment in severe cases, according to the national protocol, involved administration of dexamethasone, remdesivir, anakinra, tocilizumab, monotherapy, or other associated treatments. In addition, these patients received anticoagulants and additional high-flow nasal oxygen. In cases of acute respiratory distress syndrome, noninvasive ventilation and endotracheal intubation were used.

## 4. Discussion

In early December 2019, 41 cases of respiratory infection were evaluated and confirmed positive by RT-PCR for COVID-19, then named “2019-nCoV” [21]. In those moments, probably no one had any idea of the impact that the new virus would produce.

Since the beginning of the pandemic, around the world, multiple cases of SARS-CoV-2 infection have been reported with various strains that have emerged and put pressure on public health system by rapidly spreading and potentially evading immune protection through vaccination [22]. Genetic mutations were acquired faster than expected because previous coronaviruses usually mutate at lower rates compared with other RNA viruses, such as influenza and HIV [23].

Three months after the appearance of the first cases in Wuhan, Romania reported on 26 February 2020 the first case of SARS-CoV-2 infection in the national territory, a 20-year-old male patient from Gorj county. This person came in close contact with a citizen from Italy who had been newly diagnosed with coronavirus and recently visited Romania [24]. From then until the hospitalization of the first case with SARS-CoV-2 infection in the “St Parascheva” Clinical Hospital of Infectious Diseases, in Iasi, was a matter of a few days. Thus, on 4 March 2020, a 71-year-old man, who had returned from Lombardy complaining of respiratory symptoms, was confirmed to have COVID-19 and became the first patient hospitalized in our clinic [25]. Starting that time and until now, more than 10,000 COVID-19 patients have required hospitalization in our hospital.

The period to which the present study refers coincides with the 4th wave of SARS-CoV-2 infection in Romania, in which the dominant spread variant was Delta (B. 1. 617. 2), with most severe forms of COVID-19 and a high death rate recorded.

In the current study, we analyzed the clinical characteristics, treatment, and evolution of patients with SARS-CoV-2 infection admitted at “St. Parascheva” Clinical Hospital of Infectious Diseases in Iasi, for a period of 5 months (between July and November 2021), including in the study all patients with a diagnosis of COVID-19 confirmed by RT-PCR. In that period, the share of moderate cases of SARS-CoV-2 infection constituted almost half (47%) of the total number of studied cases. According to a large study from China, more than 80% of confirmed cases of coronavirus are not severe forms. However, even such a long persistence of symptoms could make the epidemic more difficult to control [26].

Statistics show that the elderly were the most susceptible category to COVID. Regarding the gender distribution, globally, there is no clear trend regarding the predisposition to infection with COVID-19. However, in the vast majority of countries, a clear pattern of mortality emerged; men appear more likely than women to die from COVID-19 once infected [27]. In the present study, the majority of those affected were over 60 years old. The female gender and people from urban areas predominated. The number of deaths recorded during the study period was 325, 165 (50.77%) being male patients.

The predominance of the urban environment may be because of large population, public transit, and crowded travel conditions. Several studies showed that the current socio-geographical context of some countries can lead to an increased number of COVID-19 cases, and this was also observed in our region. Tourism was the main cause involved in SARS-CoV-2 infection spreading, therefore travelers developed a higher risk of infection [28].

COVID-19 can affect anyone, young or adult, with or without associated pathologies. However, advanced age and the presence of associated diseases may represent major factors in developing of severe stages and for increasing death rates [29,30,31]. According to our study, severe infection was observed in older patients with COVID-19 and in 1242 (71.71%) cases was found at least one associated disease.

Respiratory impairment, represented by cough and febrile episodes, was encountered in most cases, while other patients presented different symptoms, such as diarrhea, anosmia, and ageusia, but in a smaller percentage. If we refer to the changes in laboratory values, the most frequent anomalies that we could observe in this study were the leukopenia and lymphopenia, as well as the increase in CRP and LDH levels, data consistent with most studies [31,32,33].

Since the start of the COVID-19 pandemic, scientists have conducted numerous studies to identify suitable therapeutic agents to treat COVID-19. Various drugs approved by the World Health Organization (WHO) and the Food and Drug Administration (FDA) were used during the pandemic. Antibiotics, antivirals, anti-inflammatories, anticoagulants, and some common medications along with combination therapies are part of the treatment or supportive care of COVID-19 patients [34]. Biological therapy (20.09%) with tocilizumab and anakinra, antivirals (56.76%) such as remdesivir and favipiravir, and new monoclonal antibody therapies (0.58%) were used as treatment for the patients from our study.

## 5. Conclusions

The novelty brought in the present study is to highlight the profile of patients and the characteristics of severe forms of COVID-19, in the northeastern region of Romania, during the 4th pandemic wave.

The epidemiological characteristics of SARS-CoV-2 infection recorded in our study were mostly the same with characteristics of COVID-19 from all over the world. We found that in our hospital, most of the cases presented a moderate form of SARS-CoV-2 infections (47%) and it was an increased number of patients with severe forms (38.80%). Mortality was increased in these patients with 18.76% deaths. Furthermore, as all around the word, treatments according to protocols were used with antivirals, monoclonal antibodies, and biological therapy.

The hospital serves the entire northeastern area of Romania, providing care and treatment for admitted patients with severe forms of COVID-19, who required monitoring and treatment in Intensive Care Unit also. During the pandemic, a new extended area dedicated to patients with SARS-CoV-2 infection was opened, thus supplementing the necessary beds, especially those with the possibility of providing additional oxygen support through concentrators and mechanical ventilators. It was a real exercise of strength that through a good collaboration and interdisciplinary involvement, we passed through the very difficult pandemic COVID-19 waves.

The identification of the most frequently affected age groups, the characteristics of the patients with SARS-CoV-2 infection, the risk of mortality among patients with associated pathologies, the predominant variant, and the need for hospitalization in intensive care departments of severe cases can raise an alarm signal regarding the protection of the categories of persons at risk. Implementation of information and awareness programs such as the need for vaccination are required. In the event of a new pandemic, the population, physicians, and researchers, therefore, will know what to expect, the latter being able to develop new therapies and new standards of care in order to respond better to patients’ needs.

## Figures and Tables

**Figure 1 jpm-12-01928-f001:**
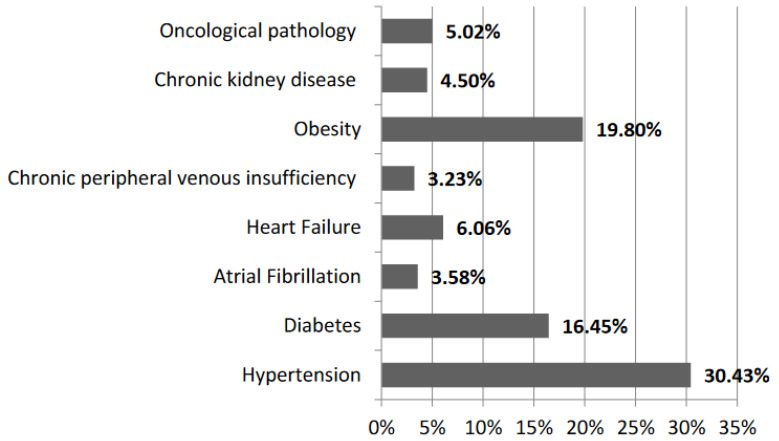
Associated pathologies in our patients.

**Figure 2 jpm-12-01928-f002:**
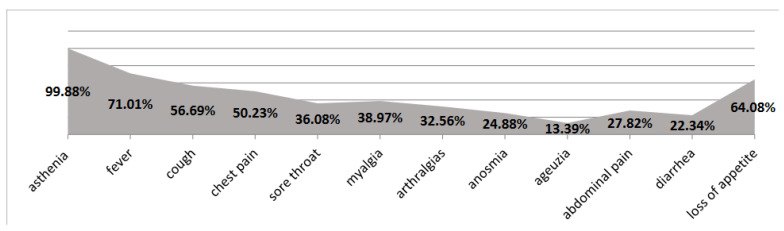
Symptoms at patients from our study.

**Figure 3 jpm-12-01928-f003:**
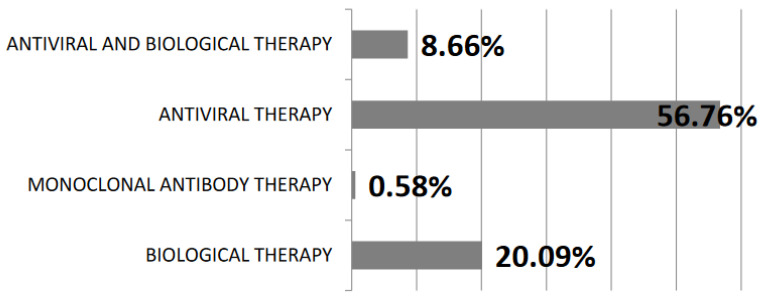
Therapies used in SARS-CoV-2 infection.

**Figure 4 jpm-12-01928-f004:**
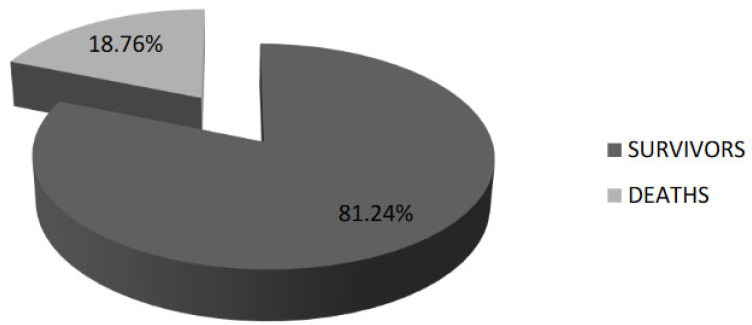
Evolution of the cases with SARS-CoV-2 infection in our study (deaths and survivors).

**Table 1 jpm-12-01928-t001:** Demographic characteristics of patients.

Characteristics		Number	%
**Total Patients**		**1732**	**100**
Age (years)	Mean	67 ± 3.4 years	
	0–14 years	52	3%
	15–24 years	29	1.6%
	25–34 years	57	3.29%
	35–44 years	157	9.06%
	45–54 years	247	14.26%
	55–64 years	323	18.65%
	65–74 years	474	27.37%
	>75 years	393	22.69%
Gender	Female	987	56.99%
	Male	745	43.01%
Residence	Urban	982	56.70%
	Rural	750	43.30%

**Table 2 jpm-12-01928-t002:** Characteristics of death cases in our study.

Characteristics		Number	%
Deaths		325	100
Age (years)	0–14 years	0	-
	15–24 years	0	-
	25–34 years	0	-
	35–44 years	3	0.92%
	45–54 years	21	6.64%
	55–64 years	54	16.61%
	65–74 years	98	30.15%
	>75 years	149	45.84%
Gender	Female	144	44.30%
	Male	181	55.70%
Residence	Urban	206	63.38%
	Rural	119	36.62%

## Data Availability

All data generated or analyzed during this study are included in this published article.

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
