# Peer review of "Characteristics and Trends of COVID-19 Infection in a Tertiary Hospital in Romania: A Retrospective Study"

_jpm, 2022, doi:10.3390/jpm12111928_

Round 1

Reviewer 1 Report

The authors present a retrospective study based on the medical records between July and November 2021. More investigation and analysis should be employed to make it for publication.

Comments:

1. The age range of underlying disease cases should be analyzed

2. For deaths, more detailed investigations should be analyzed such as age range, treatment, underlying diseases, etc.

3. comma is used in percentage numbers such as (78,46%) in line 27, line 109, Figure 2, Figure 3, line 145-146 and line 219.

4. The dominant spreading variant during that period is Delta. Can the authors confirm that?

Reviewer 2 Report

This is a very neat descriptive study with clear message presented. However, the conclusions they got seem to have very limited significance and interest to the readers at the current time point. To improve the manuscript, just a few comments and questions. 

The rounding of digits should be consistent throughout the whole manuscript.

Is there any particular reason for choosing this time window ( 07/2021-11/2021)? You may want to comment a bit in the manuscript, otherwise readers may wonder if there is any selection bias issues here. 

You may want to highlight some novel findings you got, potentially? And is there any limitations of your study? 

What are the potential implication of you study at the policy or intervention level when a new pandemic comes? 

Round 2

Reviewer 1 Report

I agree the manuscript has been sufficiently improved to warrant publication in JPM.

Reviewer 2 Report

The authors have addressed majority of my concerns and thank you for your revision.